# Socioeconomic and Environmental Benefits of Expanding Urban Green Areas: A Joint Application of i-Tree and LCA Approaches

**Mariana Oliveira** [1] , **Remo Santagata** [2,*] , **Serena Kaiser** [1] , **Yanxin Liu** [3] , **Chiara Vassillo** [2] ,
**Patrizia Ghisellini** [2] , **Gengyuan Liu** [4] and **Sergio Ulgiati** [4,5]

1   International PhD Programme/UNESCO Chair "Environment, Resources and Sustainable Development",
    Department of Science and Technology, Parthenope University of Naples, Centro Direzionale, Isola C4,
    80143 Naples, Italy
2   Department of Engineering, Parthenope University of Naples, Centro Direzionale, Isola C4,
    80143 Naples, Italy
3   School of Management and Engineering, Capital University of Economics and Business, Beijing 100070, China
4   State Key Joint Laboratory of Environment Simulation and Pollution Control, School of Environment, Beijing
    Normal University, Beijing 100875, China
5   Department of Science and Technology, Parthenope University of Naples, Centro Direzionale, Isola C4,
    80143 Naples, Italy
*   Correspondence: remo.santagata@assegnista.uniparthenope.it

**Abstract:** Green infrastructures deliver countless functions for counteracting climate change, air pollution, floods, and heat islands, contributing at the same time to water and carbon recycling as well as to renewable energies and feedstock provisioning. Properly addressing such environmental problems would require huge investments that could be decreased thanks to the further implementation of urban forests. Local administrations are designing participative projects to improve territories and their living conditions. The i-Tree Canopy modelling tool and the life cycle assessment method are jointly applied to evaluate the potential benefits of increasing tree coverage within the boundaries of the Metropolitan City of Naples, Southern Italy. Results highlighted that tree coverage could increase by about 2.4 million trees, thus generating 51% more benefits in pollutants removal, carbon sequestration and stormwater management. The benefits are also explored and confirmed by means of the life cycle assessment method. The potential tree cover is expected to provide a total annual economic benefit of USD 55 million, purchasing power parity value adjusted, representing USD 18 per citizen and USD 99,117 per square kilometre of implemented urban forest. These results can support a potential replication elsewhere and provide a reference for the sustainable improvement of cities by expanding urban green areas.

**Keywords:** urban forest; green infrastructures; i-Tree tool; life cycle assessment; ecosystem functions; nature shaped circularity

## 1. Introduction

In the last decades, the dramatic rise of urbanization rates and consequent degradation of urban environments has directed society's attention towards the natural environment, urban forests ecosystems and other urban green infrastructures [1–5]. Such attention has increased awareness about the role of urban trees and green spaces as nature-based solutions [6,7] to improve the quality of life and wellbeing of the current generations and those to come [8], creating more resilient cities [6,9,10] in contrast to the pressing urban challenges [11]. Proper planning, management, and conservation in policy agendas represent conditions for maximizing their beneficial role [11,12] and making cities more resilient to future shocks [3].

Wellbeing depends on the local geography, culture, and ecological circumstances, being affected by the availability of basic materials for a good life and the presence of healthcare services, job opportunities, and relationships within urban communities [13]. Therefore, urban green spaces can fulfil some specific functions and address other global and local issues, such as climate change mitigation and adaptation, food dependency and flood protection, high standards of health and wellbeing, employment, and income needs [3,10,14–17]. Urban forests also deliver ecosystem functions related to carbon storage/sequestration, air quality, stormwater management, energy, habitat, noise, and microclimate. Most of these functions translate into social, economic, health, visual, and aesthetic benefits for urban dwellers [5]. Moreover, urban agriculture improves the quantity and quality of food, providing higher income and employment and promoting community development by intensifying social relationships capable of breaking down cross-generational barriers and distances [18,19].

Linear economy models and categories can mainly describe the monetary transactions related to these relationships. On the contrary, the concept of commons arises by focusing on the benefits of urban reforestation measured in terms of non-profit-oriented categories. The main aspect of commons is their irreducibility to private and public property and the capability to provide socioeconomic and territorial cohesion. However, besides the need for a collective perspective aiming at protection and preservation, in many situations, green areas belong to public institutions or private owners. Hence, civil society should abandon the owner-based perspective in favour of a rights-based active involvement, putting the collective dimension in a central position [20]. The collective dimension represents the physical and theoretical space to create and utilize commons, and the development of human societies creates new commons.

Consequently, the political struggle created by social movements represents a perfect field for shaping and growing commons since their substance is not ontological but originated from their relevance in specific contexts [21], which explains why commons should always be free from profit [22]. The boundaries of commons are often identified with the boundaries of natural goods. Moreover, commons can be born and enjoyed in urban environments, since a more comprehensive set of new commons is recognized within urban frameworks [23]. Beyond the dichotomy between natural and urban commons concepts, green areas are desirable and should be included in urban environments: On the one hand, cities represent aggression against the environment and ecosystems. On the other, urban residents present an urgent need to interact with nature since perceived as an essential need for health and wellbeing. Indeed, the growing contact with nature represents an increment of environmental awareness in people's lives [24,25].

At the global level, several countries (e.g., China, Ghana, Ethiopia) and cities (e.g., New York, Melbourne, and Berlin) have developed urban forestry projects to redesign their forestry policies and increase their green stocks [3,26]. In Italy, the government approved the "Climate Bill", aiming at expanding the national forest heritage, among other actions [27]. Additionally, several cities (e.g., Milan, Modena, Ferrara, Prato, Naples) are planning urban trees expansion programs [12]. In 2013, Italy established Law n. 10 "Standards for the development of public parks and gardens" to define criteria and guidelines for creating green multifunctional systems. This national urban green strategy is divided into three theme areas: biodiversity and ecosystem services, climate change and heat islands, wellbeing, and quality of life.

Consequently, the Campania Region (southern Italy) also focused on developing urban green areas. Regional Law n. 17, "Establishment of the system of regional interest urban parks", defines the urban green system as a set of spaces with environmental and landscape value or of strategic importance for the ecological balance in territorial contexts with high anthropic impact. In 2019, the Metropolitan City of Naples (MCN), in the Campania region, released the "*Ossigeno Bene Comune*" project (OBC Project)—"Oxygen as a Common" in English—to counteract climate change, excessive urbanization and land consumption and

regenerate the MCN landscape through nature-based solutions, by planting three million trees—one per inhabitant—throughout its territory [28].

Another important aspect to be addressed when dealing with contemporary urban systems is what we can name "nature-shaped circularity" or "nature-based circular economy pattern". Circular economy (CE) is an evolving concept embodying internal complexities and multiple definitions. As pointed out by many authors [29–33], CE is an economic framework aiming to minimize resource use and waste generation by making the most out of available resources. The "nature-shaped circularity" pattern (through ecosystem services) provides a way forward to operationalize CE designing for green infrastructure planning. In so doing, it enhances the impact of CE on policies and related practices. For instance, accounting for the ecosystem services of urban green infrastructures can (i) provide a more accurate picture of the composition of the urban energy mix (including renewable energy provision from local biomass), (ii) reveal the impacts of green infrastructure on the amount of energy use (including mitigation of energy demand in buildings), and (iii) affect the dynamics of biogeochemical processes in cities (microclimate regulation and carbon sequestration by plants and soils) [34]. Thus, a circular urban system (including resource input, waste generation, emissions, and energy leakage) can be redesigned by nature-based slowing, closing, and narrowing energy and material loops, which represents a change of paradigm towards effective circularity.

Sustainable city development corresponds to a rise in the quality and quantity of all stocks considered as a source of wealth for all countries: natural, cultural, human, and manufactured [35–37], moving away from the fossil fuels-based society and linear economy towards renewable sources of energy and circular economy [30,38,39]. Therefore, besides monitoring the economic and financial assets and their flows, such as GDP [37,40], monitoring the ecosystem services stocks and flows as natural capital assets is also mandatory. However, evaluating the ecosystem functions flows is a challenge since every individual can enjoy them without any monetary payment: this suggests that the evaluation of ecosystem services should not be performed by employing a capital-oriented analysis [22]. From this perspective, ecosystem functions can be classified as commons, according to the definition given above.

This study aims to investigate and discuss the environmental, social, and economic benefits delivered by the increased presence of trees in a territory and to its inhabitants. The current and the potential tree coverage scenarios and the related benefits are assessed by the integration of the i-Tree Canopy online tool [41] and the life cycle assessment [42,43] method to quantify the ecosystem functions, including an economic perspective and pollution sequestration, to support future policies for urban green areas expansion projects and investments. The i-Tree Canopy tool is fairly used within the scientific literature to assess not just tree canopy cover but also other cover classes, thanks to its efficiency in making land cover assessments relatively easy by using aerial imagery [44–48]. However, very little seems to be present about using the i-Tree Canopy tool together with the LCA method [49], and the simultaneous use of the two frameworks to quantify ecosystem functions and assess the pros and cons of reforestation programs seems to be absent.

## 2. Materials and Methods

### 2.1. The Study Area

The Metropolitan City of Naples (MCN) in the Campania region, Southern Italy, represents the former Province of Naples, divided into 92 municipalities (Figure 1). The Municipality of Naples is the administrative centre. The MCN covers about 1200 km$^2$ and has a population of about 3 million inhabitants. The study area includes some of the highest density of population among Italian municipalities (e.g., Portici and Casavatore have about 12,000 inhabitants/km$^2$), two volcanic sites (Vesuvius and Phlegraean Fields), a long coastline, and urban areas that extend without interruption also across agricultural fields and green areas, thus presenting several types of land cover. The area falls under a Mediterranean climate zone, therefore characterized by Mediterranean vegetation, with

diversified agrarian production (potato, peach, apricot, tomato, fennel, plum, grapes, tomato, cauliflower, broccoli, and strawberry) [50].

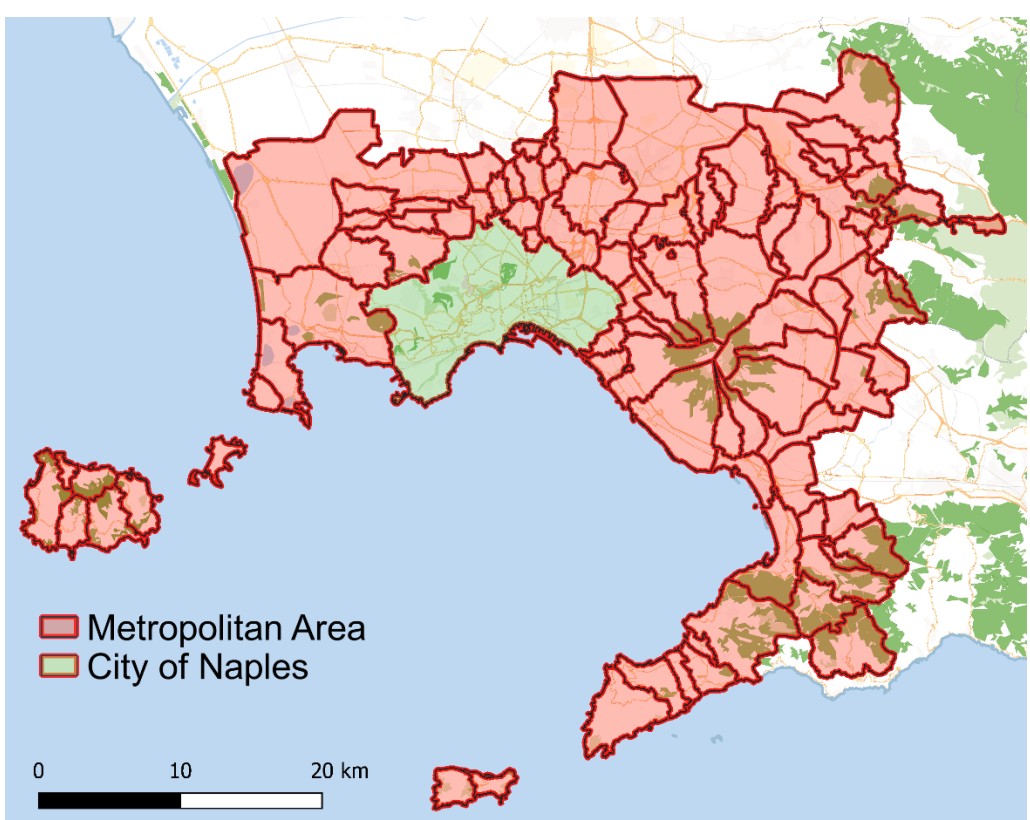

**Figure 1.** The study area of the Metropolitan City of Naples. The red borders highlight the 91 municipalities of the former Province of Naples (in red) and the City of Naples (in green).

The area accounts for three state-owned forests (Phlegraean Area, Mount Cumae's and Roccarainola's forests), which cover more than 1000 ha mainly with elm, elder, hawthorn, ash, chestnut, alder, and beech tree species [51]. The municipality of Naples includes 53 different public and private parks and green areas, almost entirely managed by municipal services, presenting a tree population of above 60,000 elements (very common *Pinus pinea* to different kinds of citrus trees, *Ginko biloba*, *Olea europaea*, *Cupressus*, and *Ficus*), with a large part of trees aged more than 50 years. Their management within the urban context represents a challenge for administrators [52]. Between 2011 and 2015, the number of new plantations exceeded the number of cuts of trees by 1.58%. However, Naples is 88th among the 105 Italian provincial chief towns, having just six trees per 100 inhabitants [53]. Due to high urbanization, the urban tree stock is minimal.

The most represented cover classes within the MCN 2018 land cover, assessed by CORINE Land Cover [54,55], are the broad-leaved forest (≈21%), the non-irrigated arable land (≈17%), the complex cultivation patterns (≈15%), the fruit trees and berry plantations (≈12%), the continuous and discontinuous urban fabric (respectively ≈12% and ≈8%), and the industrial or commercial units (≈2.5%)—confirming the strong agricultural and naturalistic tendency of the investigated area (65% of the territory).

## 2.2. i-Tree Canopy

The land cover assessment of the MCN area was performed using i-Tree Canopy tool v.7, using a classification method based on automatically retrieved aerial imagery. The tool is part of the i-Tree Tools software suite that provides urban and rural forestry analysis and benefits assessment tools. The i-Tree Canopy tool estimates tree coverage, as

well as other user-decided cover classes, by generating random points within a specified area to be manually classified [41]. In addition, it allows the estimation of the coverage area by providing the most recent aerial Google Maps/Google Earth imagery data (with a resolution of about 0.3 m) in order to perform a statistical analysis of the classification of randomly generated points [48].

The i-Tree Canopy analysis is performed following three steps:

1. Drawing boundaries or importing a file with the boundaries of the investigated area. Boundaries for the MCN were retrieved from a dedicated repository provided by the administration (http://sit.cittametropolitana.na.it/, accessed on 27 August 2019);
2. Naming the cover classes to be assessed. In this work, the cover classes are defined encompassing all the different land cover types present in the studied area (Table 1).
3. Classifying points into the defined cover classes.

**Table 1.** Cover classes used for the land cover assessment of the Metropolitan City of Naples.

| Item | Cover Class | Description |
|------|-------------|-------------|
| TI | Tree over impervious | Trees over impervious ground |
| TP | Tree over pervious | Trees over pervious ground |
| AG | Agricultural land | Pastures, crops or fallows |
| GH | Grass/herbaceous cover | Yards, parks or fields |
| IN | Impervious non-plantable | Roads, rails, roofs or monuments |
| PN | Pervious non-plantable | Dirt road |
| IP | Impervious partially plantable | Sidewalks, parking lots or plazas |
| PP | Pervious partially plantable | Bare earth |
| SB | Shrubs/bushes | Stem based vegetation smaller than trees |
| WT | Water | Ocean, estuary, river, lake, wetland, etc |
| OT | Other | Other surfaces |

In Table 1, the current tree coverage is represented by TI and TP, and the potential tree coverage is represented by IP (e.g., sidewalks, parking lots, or plazas) and PP (e.g., bare earth).

The accuracy of the final assessment will depend upon the ability of the analyst to correctly classify each randomly generated point into one of the chosen cover classes. The precision of the estimate increases with the increase in assessed points, as the standard error (SE) will decrease. Being "$n$" the number of points assessed in each class and "$N$" the total number of assessed points, *SE* for each class is calculated as in (1):

$$SE = \sqrt{\left(\frac{pq}{N}\right)},\tag{1}$$

where $p = n/N$ and $q = 1 - p$.

If $n < 10$, the *SE* is calculated as (2):

$$SE = \left(\sqrt{n}\right)/N,\tag{2}$$

Endreny et al. (2017) [48] state that 500 random points are adequate to survey megacities. In this study, to produce more accurate results, 803 points were assessed and classified according to the user-defined cover classes reported in Table 1.

The benefits investigated with the i-Tree Canopy tool are based on the percentage of tree coverage and climatic conditions. Table 2 reports the conversion factors for uptaken mass of different polluting flows and for volume of avoided runoff per unit area of trees per year, and the related economic values. The removal rates and monetary values of the considered environmental functions are derived from analyses conducted in the United States using i-Tree Eco, a component tool of the i-Tree software suite, within urban and rural areas and then aggregated at the national level [56]; data are then provided as specific sets related to chosen locations, within the report of tool results. Monetary values for pollutant removal are estimated as the incidence of adverse health effects resulting from changes

in pollutants concentrations [56]. The i-Tree Canopy tool, as well as the entire i-Tree suite, is developed by the USDA Forest Service and the Davey Tree Expert company, among others, focusing on USA territories. Thus, the average trees benefits values are calculated, regardless of tree species, based on climatic conditions for USA territories only (and in more recent times for the United Kingdom and Sweden too). In order to apply the tool to regions outside the ones included in the suite, a solution would be to analyse the annual trends and distribution of temperature and precipitations of the investigated location and find the most similar ones in similar climatic areas within the United States. In this study, to overcome geographical data limitation, the Horry County in South Carolina, USA, was selected based on similar MCN climate conditions.

**Table 2.** Removal rate and monetary value of benefits of tree coverage estimated by i-Tree Canopy.

| \multicolumn{4}{c}{**Annual Air Pollution Removal Benefits**} | | | |
|---|---|---|---|
| **Item** | **Description** | **Removal Rate (g/m$^2$/yr)** | **Monetary Value (USD/t/yr)** |
| CO | Carbon monoxide | 0.03 | 192.29 |
| NO$_2$ | Nitrogen dioxide | 0.22 | 63.03 |
| O$_3$ | Ozone removed | 7.37 | 549.10 |
| PM$_{10}$ | Particulate matter greater than 2.5 microns and less than 10 microns | 1.47 | 889.02 |
| PM$_{2.5}$ | Particulate matter less than 2.5 microns | 0.05 | 38,529.50 |
| SO$_2$ | Sulphur dioxide | 0.19 | 23.54 |
| \multicolumn{4}{c}{**Annual Hydrological Benefits**} | | | |
| **Item** | **Description** | **Tree effects (L/m$^2$/yr)** | **Monetary Value (USD/m$^3$/yr)** |
| AVRO | Avoided runoff | 0.60 | 2.60 |
| E | Evaporation | 14.94 | |
| I | Interception | 14.95 | |
| T | Transpiration | 107.41 | |
| PE | Potential evaporation | 545.19 | |
| PET | Potential evapotranspiration | 488.30 | |
| \multicolumn{4}{c}{**Carbon Benefits**} | | | |
| **Description** | | **Carbon Rate (t/ha/yr)** | **Monetary Value (USD/t)** |
| Carbon sequestered annually | | 30.60 | 187.99 |
| Carbon stored in trees' lifetime | | 768.48 | |

### 2.3. Life Cycle Assessment

The life cycle assessment method, standardized by ISO standards and ILCD handbook [42,43,57], assesses the potential environmental burdens of human-dominated processes and systems in a "cradle to grave" perspective, from the extraction of raw materials to the disposal of generated waste, through distribution and use. The results provided different kinds of impact categories, including emissions and resource consumption [58]. It is performed by following a four-step procedure:

(1) Goal and scope definition: in this phase, the objective, the functional unit (FU) and the burdens of the investigated case study are clearly defined. In this work, the LCA method is used to support and integrate the results of the performed i-Tree Canopy study. The chosen FU is the assessment of the impact categories affected by the ecological functions delivered by the current and potential tree cover within the boundaries of the Metropolitan City of Naples.

(2) Inventory analysis: LCA analyses are performed by means of specific inventories listing all relevant input and output flows enabling processes and/or subprocesses within the investigated case studies. Inventories are usually dimensioned considering the chosen FU, and include primary data (directly collected), secondary data (from scientific literature, databases, etc.), and tertiary data (calculations and assumptions). In this work, data for the used inventories come from the i-Tree Canopy tool, listing the pollution and hydrological benefits of tree cover. Hence, the inventory is built considering the annual mass of uptaken flows: particulate <2.5 µm; particulate >2.5 µm and <10 µm; sulphur dioxide; ozone; nitrogen dioxide; carbon monoxide and carbon dioxide; and the annual avoided water runoff in the assessed area in the present and future potential scenarios.

(3) Impact assessment: this step translates the input and output flows in the inventories into potential impacts in different categories by means of specific characterization factors related to distinct impact methods. In this work, the avoided pollution and hydrological effects of the current and potential tree cover are translated into characterized impacts by using the SimaPro software v.9.1.1.1 (https://network.simapro.com, accessed on 15 September 2022), the Ecoinvent database v.3.6 [59], and the ReCiPe midpoint (H) method v.1.04 [60]. The software, the database and the impact method work together to classify the burdens deriving from the assessed inventory into impacts related to specific categories. The various emissions in the different compartments and resource use are then characterized by means of particular factors to express them into the proper units.

(4) Results interpretation: in this phase, the results are carefully checked and evaluated to understand the characteristics of the investigated case study, propose solutions, find the hot spots, etc.

## 3. Results

This work explored the benefits deriving from the present and the potential tree coverage within the area of the MCN. Among the investigated cover classes, tree over impervious (TI) and tree over pervious (TP) represent the current tree coverage characterized by trees whose canopies stand above cemented sidewalks and plazas and bare earth and meadows, respectively. The potential tree coverage classes impervious partially plantable (IP) and pervious partially plantable (PP) were added to the potential scenario, considering both impervious and pervious soils where trees could be planted without impairing human activities (i.e., parking lots, sidewalks, parks, marginal lands, etc.).

### 3.1. i-Tree Canopy Results

The MCN cover area, estimated by means of the assessed 803 points, is mainly represented by the 29% trees planted over pervious grounds (TP), 23% impervious area where trees are impractical (IN) and 17% agricultural land (AG) (Table 3).

The area currently covered by trees within the MCN boundaries was identified in 251 points (TP and TI) and was estimated at 367.26 square kilometres. The area for the potential planting of new trees was identified in 128 points (IP and PP), equal to about 187.28 square kilometres (16% of the entire MCN area), which represents a potential increment of 51% in green spaces. The estimated area needed to plant one million is 80 square kilometres of forests [61]. Thus, around 240 square kilometres would be required to achieve the goal of planting three million trees, which is 30% larger than the identified available area. The analysis performed in this work suggests a current availability for plantations equal to around 2.34 million trees.

**Table 3.** Number of points assessed and estimated area for each cover class.

| Cover Class | N° of Points | Area (km²) | ±SE (km²) | % Cover |
|---|---|---|---|---|
| TP | 230 | 336.53 | 29.08 | 29% |
| IN | 187 | 273.61 | 17.50 | 23% |
| AG | 136 | 198.99 | 15.51 | 17% |
| IP | 70 | 102.42 | 11.75 | 9% |
| PP | 58 | 84.86 | 10.70 | 7% |
| SB | 46 | 67.31 | 9.63 | 6% |
| GH | 30 | 43.90 | 7.86 | 4% |
| TI | 21 | 30.73 | 9.64 | 3% |
| PN | 13 | 19.02 | 5.28 | 2% |
| WT | 9 | 13.17 | 4.35 | 1% |
| OT | 3 | 4.39 | 2.61 | 0% |
| Total Area | 803 | 1174.93 | 30 | 100% |

The i-Tree Canopy tool considers a linear correlation between the presence of trees and the functions they provide. In so doing, considering that all available areas will actually be used for tree plantation, the potential scenario presents an estimated 51% increment in terms of benefits related to air pollution absorption (Figure 2a) and carbon sequestrated and stored during trees' lifetime (Figure 2b). A 51% of increment is also estimated for avoided runoff (113 megalitres of water per year) and the hydrological cycle (219,273 megalitres of water per year) of the photosynthesis steps (evaporation, interception, transpiration, potential evaporation, and potential evapotranspiration) (Table 4).

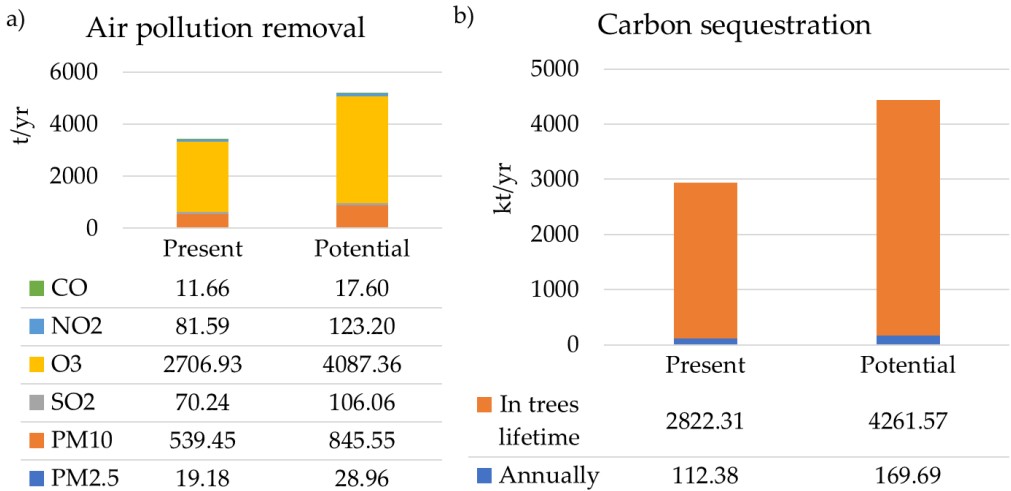

**Figure 2.** Tree benefits: (**a**) amount of air pollution absorbed expressed in carbon monoxide (CO), nitrogen dioxide (NO₂), Ozone (O₃), particulate matter (PM10 and PM2.5) and sulfuric dioxide (SO₂); (**b**) carbon annually sequestrated and carbon stored in trees' lifetime.

**Table 4.** Hydrological benefits from tree coverage in megalitres (Ml).

| Benefits | Amount (Ml) | |
|---|---|---|
| | Present | Potential |
| AVRO | 221.81 | 334.92 |
| E | 5485.22 | 8282.46 |
| I | 5491.00 | 8291.20 |
| T | 39,447.51 | 59,564.16 |
| PE | 200,226.75 | 302,334.42 |
| PET | 179,330.20 | 270,781.45 |
| Total | 430,202.49 | 649,588.61 |

The increment of the annual economic benefits is more than USD 12.4 million per year, going from almost USD 24.5 million to nearly USD 36.9 million (Table 5); 87% of this value is due to the carbon sequestrated annually by trees. Considering the amount of carbon stored in trees during their lifetime (4261 kt), the final economic value increases by more than USD 270 million, from about USD 530 million in the present scenario to more than USD 800 million in the potential scenario.

**Table 5.** Hydrological benefits from tree coverage in Megalitres (Ml).

| Benefit | Description | Value (USD) | |
| --- | --- | --- | --- |
| | | **Present** | **Potential** |
| Carbon sequestration | Sequestered annually in trees | 21,126,481 | 31,900,144 |
| Air pollution removal | CO | 2242 | 3385 |
| | NO$_2$ | 5143 | 7766 |
| | O$_3$ | 1,486,373 | 2,244,364 |
| | SO$_2$ | 1654 | 2497 |
| | PM10 | 479,582 | 724,149 |
| | PM2.5 | 739,064 | 1,115,957 |
| Hydrological | AVRO | 577,302 | 871,703 |
| **Total** | | 24,417,841 | 36,869,965 |

*3.2. LCA Analysis*

The assessed environmental benefits provided by the current and the potential tree cover are configured as an annual sequestration/absorption of different potentially harmful substances in the atmosphere and as avoided water runoff that could cause hydrological damages. These benefits can be translated into LCA-avoided impacts, expressing the annual avoided repercussion in different, specific environmental sectors, thanks to the presence of trees delivering their functions. The LCA analysis of the ecological functions provided by the present and the potential assessed tree cover is reported in Table 6. The different air pollutant substances sequestered by trees and the avoided runoff in the two scenarios affect six ReCiPe midpoint (H) impact categories: global warming; ozone formation, human health; fine particulate matter formation, ozone formation, terrestrial ecosystems; terrestrial acidification; water consumption. The results confirm once again the linear correlation between environmental benefits and the presence of trees proposed by the i-Tree Canopy tool.

**Table 6.** Characterized ReCiPe midpoint impacts avoided by the present and the possible tree coverage in the Metropolitan City of Naples.

| Impact Category | Unit | Present | Potential |
| --- | --- | --- | --- |
| Global warming | kg CO$_{2\,eq}$ | $1.12 \times 10^8$ | $1.70 \times 10^8$ |
| Ozone formation, human health | kg NO$_{x\,eq}$ | $8.16 \times 10^4$ | $1.23 \times 10^5$ |
| Fine particulate matter formation | kg PM$_{2.5\,eq}$ | $4.85 \times 10^4$ | $7.33 \times 10^4$ |
| Ozone formation, terrestrial ecosystems | kg NO$_{x\,eq}$ | $8.16 \times 10^4$ | $1.23 \times 10^5$ |
| Terrestrial acidification | kg SO$_{2\,eq}$ | $9.96 \times 10^4$ | $1.50 \times 10^5$ |
| Water consumption | m$^3$ | $2.22 \times 10^5$ | $3.35 \times 10^5$ |

Planting new trees in urban environments presents both economic and environmental costs due to the cultivation of tree seedlings in specific nurseries, involving energy, chemicals, and machinery, among other input flows. A careful accounting of the mentioned environmental burdens against the analysis of benefits coming from the presence of trees could facilitate afforestation projects and urban green management plans. Table 7 shows the

ReCiPe midpoint impacts of the production and plantation of 2.34 million tree seedlings, according to the Ecoinvent database used in this study. The comparison between the numbers in Tables 6 and 7 suggests that the impacts of tree production and planting would be entirely compensated by the functions provided by these new trees in a little more than half a year ($\approx$200 days).

**Table 7.** Characterized ReCiPe midpoint impacts for growing and planting 2.34 million tree seedlings.

| Impact Category | Unit | Growing + Planting |
|---|---|---|
| Global warming | kg $CO_{2\ eq}$ | $3.36 \times 10^6$ |
| Stratospheric ozone depletion | kg $CFC11_{\ eq}$ | $1.98 \times 10^0$ |
| Ionizing radiation | kBq $Co\text{-}60_{\ eq}$ | $1.40 \times 10^5$ |
| Ozone formation, human health | kg $NO_{x\ eq}$ | $2.23 \times 10^4$ |
| Fine particulate matter formation | kg $PM_{2.5\ eq}$ | $8.29 \times 10^3$ |
| Ozone formation, terrestrial ecosystems | kg $NO_{x\ eq}$ | $2.28 \times 10^4$ |
| Terrestrial acidification | kg $SO_{2\ eq}$ | $1.57 \times 10^4$ |
| Freshwater eutrophication | kg $P_{\ eq}$ | $9.03 \times 10^2$ |
| Marine eutrophication | kg $N_{\ eq}$ | $9.27 \times 10^1$ |
| Terrestrial ecotoxicity | kg 1,4-DCB | $1.73 \times 10^7$ |
| Freshwater ecotoxicity | kg 1,4-DCB | $2.65 \times 10^5$ |
| Marine ecotoxicity | kg 1,4-DCB | $3.43 \times 10^5$ |
| Human carcinogenic toxicity | kg 1,4-DCB | $2.02 \times 10^5$ |
| Human non-carcinogenic toxicity | kg 1,4-DCB | $1.22 \times 10^7$ |
| Land use | $m^2$ a crop $_{eq}$ | $7.68 \times 10^6$ |
| Mineral resource scarcity | kg $Cu_{\ eq}$ | $5.50 \times 10^4$ |
| Fossil resource scarcity | kg $oil_{\ eq}$ | $9.44 \times 10^5$ |
| Water consumption | $m^3$ | $1.47 \times 10^4$ |

## 4. Discussion

The economic benefit of the current tree coverage, calculated by the i-Tree Canopy tool, is estimated at about USD 24 million annually ($\approx$USD 8 per citizen of the MCN per year). If the assessed potentially plantable area was used for tree plantations, the total benefits would be equal to $\approx$USD 37 million/year ($\approx$USD 12 per citizen). At the urban level, further economic savings that are not currently included within the perspective of the tool, which, as seen, operated a linear correlation among tree cover and ecological functions provided, would also be found. For instance, tree shadings would affect the local temperature, inducing a reduction in the expenses for electricity that would vary from USD 4 to USD 166 per tree annually due to reduced use of cooling and heating systems [62].

In order to undertake the OBC Project, which originated the idea behind the research presented in this work, the MCN administration committed to funding USD 969 million, within which planting trees is only one of the actions planned.

The total annual savings of the present and potential trees land coverage are estimated at $\approx$USD 55 million/year purchasing power parity value (PPP) adjusted (considering the value of 0.6708 for Italy in 2019 [63]). It is equivalent to $\approx$USD 18 per citizen of the MCN per year and $\approx$USD 99,117 per square kilometre of tree coverage. Endreny et al. (2017) investigated 10 of the biggest megacities worldwide, with an average of $1.8 \times 10^7$ citizens, including among the tree benefits also the cost of the avoided kWh of electricity and the avoided Mbtu of energy. The median value of the additional tree coverage in the investigated megacities was estimated at 17.8%, providing an average USD 967,000/km$^2$ of tree coverage and USD 32/capita, PPP adjusted. Rogers et al. (2015) assessed the tree population for the Greater London Area, calculating an average benefit of about USD 891,000/km$^2$ and USD 23/capita (PPP adjusted) for a canopy cover equal to 21% of the territory [64].

Scaling up the economic benefits calculated in this work to the projected number of three million trees planted, it would be reasonable to expect a payback time reduced to 24 years from the entire tree coverage within the investigated area, considering the

total investment by the MCN administration and disregarding other benefits provided from the presence of urban forests. Therefore, the payback time could be further reduced in a more accurate and detailed evaluation. Further studies should include logistics and implementation costs, project duration, and job opportunities creation to provide a comprehensive evaluation of the investments and provide a complete management report.

The three million trees objective would require the implementation of significant actions to be fulfilled. Among these, the plantation of trees within the areas assessed in this work as IP—impervious partially plantable—might represent a challenge, as this cover class is largely characterized as spaces within the highly populated urban areas of the MCN (sidewalks, plazas, and parking lots represent 9% of the territory—around 102 km$^2$). In these places, trees are planted individually and spaced, improving the population approach to some ecological functions, such as local temperature and humidity control, runoff, infiltration, and flooding regulation, and creating areas for recreation and leisure. However, this might represent a significant economic cost for the city administration due to the regular maintenance required by street trees (pruning, irrigation, crown thinning, and removal). Even more efforts would be required to convert a significant fraction of the land cover assessed as "not plantable" (e.g., industrial areas, marginal lands, unused deposits, etc.) into entire "urban forests", freeing the soil from old cement and giving back life to the ground underneath. In this case, the bare ground would become reforested wood, offering tree benefits on a regional scale. This strategy also includes other benefits such as aquifer recharge, soil formation, fertility and biodiversity maintenance, recreative and touristic places creation [65], and wellbeing. However, this would require a huge initial effort from the metropolitan city administration, compensated by lesser needs on management and maintenance compared to single-planted street trees.

Nastran et al. (2022) analysed the perceived connections between ecosystem services and green infrastructures in urban ecosystems and their impacts on wellbeing, suggesting that urban forests are the most influencing among the types of infrastructures considered [66]. Valeri et al. (2021) investigated the significance of the selection of target plant species in peri-urban and agricultural areas, as well as in many areas of the MCN [67]. Evans et al. (2022) reviewed scientific literature about green infrastructures and services delivery, in particular describing how their delivery is partly modulated by the kind of spaces where they are assessed [68]. Shao and Kim (2022) investigated the urban heat islands mitigation potential of green infrastructures, at the same time dealing with climate change and providing different functions promoting sustainable development and wellbeing in urban systems [69]. García-Pardo et al. (2022) reviewed remote sensing techniques for ecosystem services analysis, pointing out the importance of the sensors used, the geographical scale and image resolution, and the need for more information and a transdisciplinary framework for the assessment of the ecosystem services [70].

The results and insights obtained in this work, together with the presented joint implementation of the i-Tree tool suite and the LCA assessment method, could be useful for administrators, policymakers, urban planners, and organizations of different natures on successfully planning and managing urban greening projects and interventions, receiving countless environmental functions and benefits in return. The application of both the i-Tree and LCA tools made it possible to analyse the specific polluting substances and hydrological features investigated in this work and characterize them into specific impact categories in different environmental sectors. This allowed a wider understanding of the investigated case study, enabling the possibility of looking at it from multiple perspectives and calculating the time needed to compensate the related environmental and economic investments. However, the i-Tree Canopy tool might adopt a too simplistic point of view when establishing only a linear correlation between the presence of trees and the provided functions. This might be true in some cases but untrue in others, where the synergies among trees and the inclusions of more functions could indicate different types of correlation (e.g., exponential). Thus, the kind of analysis performed in this work could benefit from a deeper and wider investigation of the ecological functions related to trees. This kind

of analysis, together with stronger efforts for the conversion of already identified areas, incentives for private sector initiatives, and ecological management for agriculture, can provide further benefits to biodiversity, crops and pastures productivity [71].

The findings of this work then suggest the importance of ecosystems functions and the need for proper planning and management in order to correctly account the balance between investments and advantages. As the kind and the number of functions can be affected by local conditions and by the viewpoint adopted in their analysis, the simultaneous application of different methods, as the performed joint application of LCA and i-Tree Canopy, can provide a very beneficial, multiple perspective to proper planning in worldwide urban environments. Moreover, there seems to be a lack in scientific literature of joint i-Tree Canopy/LCA applications, reinforcing the novelty of the presented work and proposing a framework for benefit/constraint analyses of designed and implemented projects for green infrastructures.

*Practical Implications: Social, Environmental, and Economic Benefits*

The results of this work show opportunities to improve people's quality of life, reduce environmental damage, and bring economic development by the provision of ecological functions. The increment of 51% in new green areas reduces 51% of harmful hydrological events, air pollution, and greenhouse gases. This is also confirmed by the avoided emission highlighted by the LCA analysis.

Greenhouse gas emissions and air pollutants are harmful to crops' productivity, human health, and, consequently, the economy [72]. Air pollutants are responsible for pulmonary diseases, cancer, and respiratory infections, increasing death rates [73] and public health costs. Additionally, the $O_3$ exposure reduces plant transpiration and heat fluxes, diminishing rainfall and increasing air temperature. All these factors negatively impact the crop yield, from 18% to 45%, representing approximately 35% of loss related to gross primary productivity (GPP) [74].

> *"The interaction with nature is a fundamental pillar for individuals' wellbeing, regardless of their geographical, cultural, or socioeconomic background"*. [24]

Many city residents have no daily contact with local nature and biodiversity, which also impacts individual wellbeing. Different studies show the health benefits of living nearby green areas, improving human wellbeing by reducing blood pressure, heart rate, and muscle tension [75,76]. Unfortunately, too many individuals are far from natural areas in their daily life, and this distance might impact their perception of natural environments and local biodiversity. However, it is also possible to notice that personal perceptions are affected by social, cultural, and psychological backgrounds that should be addressed and investigated in local communities.

The *New York Times* newspaper, on 21 September 1966, reported the former US President Ronald Reagan's words talking about trees as sources of raw materials: "A tree's a tree. How many more redwoods do you need to look at? If you've seen one, you've seen them all". In 1973, Martin Krieger published an article in *Nature* questioning the utility of urban trees: the use of plastic trees could replicate the recreational and shading functions of natural ones; he argued that the definition of natural environment depends on culture and society, and that "preservationism" is more expensive than pragmatic conservationism [77]. The argument resurfaced in 2007 when the journal *Nature* published a blog post discussing Krieger's idea, concluding that living in a Disneyfied world costs more than the direct monetary value of fake wilderness, being the actual cost is the loss of ecosystem value [78].

Nature and urban green spaces as elements of the urban landscape are fundamental to improving the quality of life of citizens. Therefore, they must be considered in urban planning policies to reflect the needs, economic possibilities, and customs of the city's inhabitants. It is, in any case, an interpretation that takes its inspiration from the hope that biopolitics can have the possibility of winning against biopower [79].

Urban nature is vital in all its manifestations, from private gardens, tree plantations, and city parks to land used for recreational purposes within or nearby the city.

## 5. Conclusions

This study aimed to evaluate the socioeconomic and environmental benefits of planting trees in urban environments. The area within the boundaries of the Metropolitan City of Naples that is available for planting new trees is enough for 2.3 million new plantations on impervious ground (sidewalks, plazas, parking lots, etc., 9% of the territory—around 102 km$^2$) and pervious ground (7% of the region—about 85 km$^2$). The i-Tree Canopy tool is a useful web tool for helping policymakers to understand the environmental and economic value of the benefits offered to society and the interest in carrying out a public project entailing urban trees, evaluating specific scenarios to meet requirements and cost-effective solutions. Its combined application with the LCA method delivers a wider understanding of the ecological sectors affected by impacts and benefits of reforestation activities. In so doing, the two instruments showed that careful planning and different strategies, considering investments, costs, and benefits in a long-term perspective, are needed. As public funds available to administrators are often very scarce, public authorities also need to understand the socioeconomic improvements of carrying out urban forestry projects compared to other alternatives. Furthermore, urban forestry projects have several positive social and cultural side-effects: the creation of new green areas for recreational and cultural meetings and the aesthetical valorisation of spaces for cultural progress and civil society involvement. However, further research and collaborations among different stakeholders and the integrated use of diverse assessment methods are needed to evaluate projects under the three pillars (environmental, economic, and social) of sustainable development and the need to accelerate the transition in the EU towards a more regenerative and circular economy at the EU level.

**Author Contributions:** M.O.: conceptualization, methodology, formal analysis, investigation, data curation, writing—original draft, writing—review and editing; R.S.: conceptualization, methodology, formal analysis, investigation, data curation, writing—original draft, writing—review & editing; S.K.: conceptualization, investigation, writing—original draft, writing—review and editing; Y.L.: conceptualization, investigation, writing—original draft; C.V.: conceptualization, investigation, writing—original draft; P.G.: conceptualization, methodology, writing- original draft; writing—review and editing; G.L.: conceptualization, writing—original draft, funding acquisition; S.U.: conceptualization, methodology, supervision, project administration, funding acquisition. All authors have read and agreed to the published version of the manuscript.

**Funding:** This study received funding from: the European Union's Horizon 2020 research and innovation programme under the Marie Skłodowska-Curie European Training Networks (grant agreement ReTraCE No. 814247), the Italian Ministry of Foreign Affairs and International Cooperation (MAECI, High Relevance Bilateral Projects, Grant No. PGR00954), the Sino-Italian Cooperation of NSFC Natural Science Foundation of China (Grant No. 71861137001), within the project "Analysis on the metabolic process of urban agglomeration and the cooperative strategy of circular economy" (2018–2020), the European Union's Horizon 2020 research and innovation programme (grant agreement JUST2CE No. 101003491), and the Fundamental Research Funds of Capital University of Economics and Business (Grant No. XRZ2022028).

**Data Availability Statement:** Data sharing not applicable.

**Conflicts of Interest:** The funders had no role in the design of the study; in the collection, analyses, or interpretation of data; in the writing of the manuscript; or in the decision to publish the results. The authors declare no conflict of interest.

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
