# Peer review of "Socioeconomic and Environmental Benefits of Expanding Urban Green Areas: A Joint Application of i-Tree and LCA Approaches"

_land, doi:10.3390/land11122106_

Round 1

Reviewer 1 Report

Ref. Review: Land - 1960415

Paper Title: Socioeconomic and environmental benefits of expanding urban green areas: a joint application of i-Tree and LCA approaches

Dear Authors and Editor,

Based on the text in the entitled paper: "Socioeconomic and environmental benefits of expanding urban green areas: a joint application of i-Tree and LCA approaches", I recommend major revisions before accepting and publishing on Land.

This study aims to investigate and discuss the environmental, social, and economic benefits delivered by the increased presence of trees in a territory and its inhabitants. The current and the potential tree coverage scenarios and the related benefits are assessed by the integration of i-Tree Canopy online tool and Life Cycle Assessment method to quantify the ecosystem functions, including an economic perspective and pollution sequestration, with the purpose of supporting future policies for urban green areas expansion projects and investments.

Please consider the following recommendations for this manuscript:

Overall text:

1)      Please revise the grammar of your manuscript. An example is shown in the description of your aims (bold text corrected).

2)      The authors explained in detail the concept of common in the introductory part; however, the section on Life Cycle Assessment should be improved. The reader cannot reproduce the methodology based on the exposed text. It is necessary to do a complementary reading from other works to understand what the authors are trying to expose in the text. Until the section on i-Tree Canopy, it was easy to follow the text. Afterwards, the text is unclear, without equations to understand the calculations and quantifications exposed in the results section, which summarised conclusions that I am unsure how the authors could conclude.

3)      Since the authors are writing to a public that does not speak Italian, please consider writing the English name of the mentioned cities if there is an English version, e.g., Milano = Milan, Napoli = Naples, etc. The same for other Italian words (plazas = squares). The mentioned names and words are at the following lines: 85, 94, 148, 149, 150, 161, 165, 172, 248, 332, 345, 384, and 462.

Introduction:

4)      Lines 74-80: “Beyond the dichotomy between natural and urban commons concepts, green areas are desirable and should be included in urban environments: on the one hand, cities represent an aggression against the natural environment and ecosystems. On the other, urban residents present an urgent need to interact with nature since perceived as an essential need for health and well-being for people. Indeed, the growing contact with nature represents an increment of the perception about the importance of the environment in environmental awareness in people’s life [23,24]”. Please consider the corrections above.

5)      Line 95: Could you please highlight “Ossigeno Bene Comune” in italic?

6)      Line 101: “Circular economy (hereafter, CE)” The word “hereafter” is not necessary.

7)      Lines 128-139: “This study aims to investigate and discuss the environmental, social, and economic benefits delivered by the increased presence of trees to a territory and its inhabitants. The current and the potential tree coverage scenarios and the related benefits are assessed by the integration of i-Tree Canopy online tool [40] and Life Cycle Assessment [41,42] method, in order to quantify the ecosystem functions, including an economic perspective and the pollution sequestration, with the purpose of supporting future policies for urban green areas expansion projects and investments. In so doing, this work aims to investigate the potential benefits, both from an environmental and from an economic perspective, of a reforestation objective at a provincial level, and the potential for integration of the LCA method and the i-Tree tool suite in providing useful insights and indicators for researchers and policymakers about possible advantages and possible constraints of local reforestation programs”.  It is suggested to remove the second part of the text since it is a repetition of the major idea exposed in the first part. Grammar corrections also are required as indicated.

Methodology:

8)      Figure 1: The study area of the Metropolitan City of Naples Napoli and its municipalities (indicated with red boards). Please consider the highlighted suggestion.

9)      I-tree Canopy: Could the evapotranspiration of the tree species in the US be different from the ones in Italy? this point should be observed and stated in the text. Based on this information you can observe carbon sequestration and the hydrological cycle, which will confirm or not your findings.

10)   Life Cycle Assessment: Please describe SimaPro software and its calculation method. The same suggestion is recommended for the ReCiPeMidpoint (H) method v.1.0.

Results:

11)   Figure 2, Table 4, Table 5: please indicate from how you got these results.

Reviewer 2 Report

The land is an international research journal, addressing a broad international audience interested in research, in order to be able to acquire new concepts and methods. The current submission does not meet these aims. First, it lacks research depth, without exceeding the limits of a case study, since the research is not framed in a theoretical context based on an extensive review of the broad international literature on this topic (specifically on LCA and I-tree). Second, its is very local in scope, analyzing the case study in a context contained to the italian borders, without attempting to draw conclusions applicable elsewhere. Third, the article presents some results, without attempting to analyze them; it looks like a report for the government, where facts and figures are presented without being analyzed. Fourth, the English is poor, with important flaws, incompatible with an international journal.

The abstract and case study starts abruptly with the description of the case study. We do no longer live in the world of geographical discoveries. Information about Napoli can be found easily on Wikipedia or elsewhere. The international readers of Land expect, based on the title, to find about the way of potential reduction of GHG emissions The authors must rethink the introduction (and the abstract) by employing a serious critical review of the broad international literature on GHG emission reduction method, and create a theoretical context for their case study. 

Similarly, the discussion and conclusions are appropriate for a report addressed to the Italian Government (be it central, regional, local), and not for a research journal. The authors should rethink the entire strategy of writing their article. It is not addressed to the italian authorities, but to an international research community. They should be able to answer questions like: what lessons would an American/English researcher learn from their manuscript? What is the take-home message for the international research community?

In this regard, the authors must pass from the stage of presenting results to analyzing them, identifying the underlying drivers and mechanisms of change, in an attempt to identify patterns present also in other case studies. Their case study must not be seen as unique, but compared with similar case studies on GHG emission using other methods of other cities; if differences exist, they should be explained and analyzed.

Some flagrant errors found on the first 20 rows, even in the title, include:
Line 149: 92 municipalities (Error! Reference 149 source not found.). 
Line 197 and 199: same as above Line 214 and 218

How did the authors get Table 2 results? reference or detailed explanation.

For LCA part:

Where is the inventory and each step of calculation which was used in Simapro?

This should be submitted with the revised version 

Reviewer 3 Report

The paper deals with the socioeconomic and environmental benefits of expanding urban green areas using two approaches. The manuscript falls within the scope of the subject matter of the journal and meets its requirements, addressing the important problem of management of urban green areas. The structure of the article is appropriate and well-ordered, the language is understandable, and the authors explain the issues discussed sufficiently. The manuscript describes applied research which has practical value, but the results and methods need to be clarified and specified properly. I propose to accept the manuscript for publication after minor revision: it seems that the Figures were prepared in a hurry, and they must be improved – Figure 1 is a bit blurry, and the text is too small. There are a lot of errors due to bibliographic references missing – lines 215, 218, 294, 295, 299 and 300. Please explain what does it mean latest Google Maps/Google Earth imagery was used to determine land cover. Please specify correctly which datasets have been used and why

Round 2

Reviewer 1 Report

The correction in this version of the manuscript attended the expectations.

Author Response

Authors thank Reviewer 1 for his efforts in improving the manuscript

Reviewer 2 Report

The revised manuscript is not revised according to my comments, The revised version ignored major comments. 

Methods:

1. Not explained the detail. 

2. Similarly, the discussion and conclusions are appropriate for a report addressed to the Italian Government (be it central, regional, local), and not for a research journal. The authors should rethink the entire strategy of writing their article. It is not addressed to the Italian authorities, but to an international research community. They should be able to answer questions like: what lessons would an American/English researcher learn from their manuscript?

3. In this regard, the authors must pass from the stage of presenting results to analyzing them, identifying the underlying drivers and mechanisms of change, in an attempt to identify patterns present also in other case studies. Their case study must not be seen as unique, but compared with similar case studies on GHG emission using other methods of other cities; if differences exist, they should be explained and analyzed.

4. How did the authors get Table 2 results? reference or detailed explanation.

5. Where is the inventory and each step of calculation that was used in Simapro? Still not provided which is one of the important things in LCA calculation. 

Author Response

From Reviewer 2:

The revised manuscript is not revised according to my comments, The revised version ignored major comments.

Authors thank Reviewer 2 for his efforts and cooperation in improving the manuscript.

Methods:

  1. Not explained the detail.

Authors are not sure of what detail the Reviewer is referring to. In the Methods section, the i-Tree Canopy tool is discussed in deep details explaining its utility and functioning (Lines 181-189), how it is operated (Lines 190-205), how it calculates results and how it accounts for errors (Lines 206-216), how the benefits are calculated per m2 of tree presence and accounting for climatic conditions (Lines 217-238).

Regarding the LCA analysis, as the Reviewer is surely well aware, it is performed throughout 4 steps, standardized by international ISOs, as stated in Lines 240-245. Then each step is described in theory and in its application to the present case study: Step 1 – Goal and Scope definition (Lines 247-253), in which the goal of the analysis and the selected functional unit are presented; Step 2 – Inventory analysis (Lines 254-265), presenting how the inventory has been built; Step 3 – Impact Assessment (Lines 266-277), explaining how results are achieved and listing the software used (Simapro v.9.1.1.1), the database used (Ecoinvent v.3.6) and the impact method for classification and characterization of impacts (ReCiPe Midpoint (H) v.1.04).

Authors think that the presented methods section is already very detailed and ask Reviewer 2 to kindly indicate what is missing in his opinion.

  1. Similarly, the discussion and conclusions are appropriate for a report addressed to the Italian Government (be it central, regional, local), and not for a research journal. The authors should rethink the entire strategy of writing their article. It is not addressed to the Italian authorities, but to an international research community. They should be able to answer questions like: what lessons would an American/English researcher learn from their manuscript?

It is not clear why the Reviewer is referring to a report for Italian authorities. As already explained, the work took the opportunity to investigate the potential for afforestation in the MCN after the metropolitan authorities issued the mentioned OBC project, but the analyses reported are not directly intended for them. The research group has an interest for and has been involved in research about green infrastructures and ecosystem functions (as mentioned in the previous rounds of reviews, some of the authors of the presented work participated in the Endreny et al. (2017) work about ecosystem services in 10 megacities).  

As clarified in Section 4: “The results and insights obtained in this work, together with the presented joint implementation of the i-Tree tool suite and the LCA assessment method, could be useful for administrators, policy makers, urban planners, and organizations of different natures on successfully planning and managing urban greening projects and interventions, receiving in return countless environmental functions and benefits. The application of both the i-Tree and LCA tools made possible to analyze the specific polluting substances and hydrological features investigated in this work and characterize them into specific impact categories in different environmental sectors. This allowed a wider understanding of the investigated case study, enabling the possibility of looking at it from multiple perspectives and also calculating the time needed to compensate the related environmental and economic investments. However, the i-Tree Canopy tool might adopt a too simplistic point of view when establishing only a linear correlation between the presence of trees and the provided functions. This might be true in some cases but untrue in others, where the synergies among trees and the inclusions of more functions could indicate different types of correlation (e.g., exponential). Thus, the kind of analysis performed in this work could benefit from a deeper and wider investigation of the ecological functions related to trees. This kind of analysis, together with stronger efforts for the conversion of already identified areas, incentives for private sector initiatives, and ecological management for agriculture, can provide further benefits to biodiversity, crops and pastures productivity [66]”.

The analysis characterized, quantified and reported the benefits of planting trees in urban environments, investigating pros and cons and comparing (economic and environmental) investments to calculate the payback time. An American/English reader, as one of any nationality, is suggested that afforestation programs are beneficial from many perspectives, with functions provided of different nature and in different amounts depending on local conditions. They are also suggested that the possibly significant economic and environmental investments are compensated in a relatively small time. They are also suggested a way of evaluating and comparing them. Also, constraints within the tools used have been identified, in order to achieve a better understanding of ecosystem functions.

The following text has been added at Line 443: The findings of this work then suggest the importance of ecosystems functions and the need for proper planning and management in order to correctly account the balance between investments and advantages. As the kind and the amount of functions can be affected by local conditions and by the viewpoint adopted in their analysis, the simultaneous application of different methods, as the performed joint application of LCA and i-Tree Canopy, can provide a very beneficial, multiple perspective to proper planning in worldwide urban environments. Moreover, there seems to be a lack in scientific literature of joint i-Tree Canopy/LCA applications, reinforcing the novelty of the presented work and proposing a framework for benefit/constraint analysis of designed and implemented projects for green infrastructures.

  1. In this regard, the authors must pass from the stage of presenting results to analyzing them, identifying the underlying drivers and mechanisms of change, in an attempt to identify patterns present also in other case studies. Their case study must not be seen as unique, but compared with similar case studies on GHG emission using other methods of other cities; if differences exist, they should be explained and analyzed.

Authors are not sure of why Reviewer 2 is referring to studies about GHG emissions when GHG emissions are never mentioned as the focus of the present case study, nor represent a prominent perspective within the theoretical implant of green infrastructures. As other works on similar case studies report, green infrastructures are beneficial in terms of the functions they provide to the urban ecosystems. These functions are related to pollution sequestration (considered within the i-Tree Canopy tool), stormwater management (considered within the i-Tree Canopy tool) and others not taken into account by the tool but mentioned in the manuscript as reduction of the heat island effect, temperature mitigation and related effect on the use of electricity in urban systems and effects on human wellbeing, among others. Relevant literature about the mentioned effects is reported and compared to the assessed case study (Comparison with Endreny et al. (2017) and with Rogers et al. (2015) in Lines 373 to 381. In section 4.1 the implications of the presence of trees are discussed in detail from a social, environmental, and economic perspective, reporting selected literature.

The Following text has been added in Line 410: Nastran et al. (2022) analyzed the perceived connections between ecosystem services and green infrastructures in urban ecosystems and their impacts on wellbeing, suggesting that urban forests are the most influencing among the types of infrastructures considered [66]. Valeri et al. (2021) investigated the significance of the selection of target plant species in peri-urban and agricultural areas, as well as in many areas of the MCN [67]. Evans et al. (2022) reviewed scientific literature about green infrastructures and services delivery, in particular describing how their delivery is partly modulated by the kind of spaces where they are assessed [68]. Shao and Kim (2022) investigated the urban heat islands mitigation potential of green infrastructures, at the same time dealing with climate change and providing different functions promoting sustainable development and wellbeing in urban systems [69]. García-Pardo et al. (2022) reviewed remote sensing techniques for ecosystem services analysis, pointing out the importance of the sensors used, the geographical scale and image resolution, and the need for more information and a transdisciplinary framework for the assessment of the ecosystem services [70]

The following references have been added to the bibliography:

Nastran, M.; Pintar, M.; Železnikar, Š.; Cvejić, R. Stakeholders’ Perceptions on the Role of Urban Green Infrastructure in Providing Ecosystem Services for Human Well-Being. Land 2022, Vol. 11, Page 299 2022, 11, 299, doi:10.3390/LAND11020299.

Valeri, S.; Zavattero, L.; Capotorti, G. Ecological Connectivity in Agricultural Green Infrastructure: Suggested Criteria for Fine Scale Assessment and Planning. Land 2021, Vol. 10, Page 807 2021, 10, 807, doi:10.3390/LAND10080807.

Evans, D.L.; Falagán, N.; Hardman, C.A.; Kourmpetli, S.; Liu, L.; Mead, B.R.; Davies, J.A.C. Ecosystem Service Delivery by Urban Agriculture and Green Infrastructure – a Systematic Review. Ecosyst Serv 2022, 54, 101405, doi:10.1016/J.ECOSER.2022.101405.

Shao, H.; Kim, G. A Comprehensive Review of Different Types of Green Infrastructure to Mitigate Urban Heat Islands: Progress, Functions, and Benefits. Land (Basel) 2022, 11, doi:10.3390/LAND11101792.

García-Pardo, K.A.; Moreno-Rangel, D.; Domínguez-Amarillo, S.; García-Chávez, J.R. Remote Sensing for the Assessment of Ecosystem Services Provided by Urban Vegetation: A Review of the Methods Applied. Urban For Urban Green 2022, 74, 127636, doi:10.1016/J.UFUG.2022.127636.

  1. How did the authors get Table 2 results? reference or detailed explanation.

Table 2 is not presenting results. As already explained, it reports the factors used by the i-Tree Canopy tool to calculate the results. The text at Line 218 has been modified as following for clarity:

Table 2 reports the conversion factors for uptaken mass of different polluting flows and for volume of avoided runoff per unit area of trees per year, and the related economic values. The removal rates and monetary values of the considered environmental functions are derived from analyses conducted in the United States using i-Tree Eco, a component tool of the i-Tree software suite, within urban and rural areas and then aggregated at the national level [56]; data are then provided as specific sets related to chosen locations, within the report of tool results.”

  1. Where is the inventory and each step of calculation that was used in Simapro? Still not provided which is one of the important things in LCA calculation.

The inventory used within LCA analyses is built according to the results showed in Figure 2 and Table 4. In order to avoid redundancy by listing the same numbers again in another table, the following text has been added to Point 2 of Subsection 2.3: “Hence, the inventory is built considering the annual mass of uptaken flows: particulate <2.5 μm; particulate >2.5 μm and <10 μm; sulfur dioxide; ozone; nitrogen dioxide; carbon monoxide and carbon dioxide; and the annual avoided water runoff in the assessed area in the present and future potential scenarios.”

Once defined an inventory, a database and an impact method, as Authors did and illustrated in the manuscript, there is nothing much of calculations going on, as they are performed within the algorithms of the software. The benefits are then compared to the impacts of growing and planting tree seedlings, that, as explained in lines 349 to 351, are retrieved from the ecoinvent database and the compensation time is calculated as a simple proportion between the impacts for these plantations and the annual recovery provided by trees.

Then, once again, Authors ask the Reviewers for indications of what is missing in his opinion.